# The Carbon Emission Intensity of Rainwater Bioretention Facilities

**Deqi Wang** [1,2]**, Xuefeng Liu** [1,2]**, Huan Li** [1,2]**, Hai Chen** [3,*]**, Xiaojuan Wang** [1,2]**, Wei Li** [4]**, Lianbao Cao** [4]**, Jianlin Liu** [4]**, Tingting Zhang** [1,2] **and Bigui Wei** [1,2,*]

[1] School of Environmental and Municipal Engineering, Lanzhou Jiaotong University, Lanzhou 730070, China
[2] Key Laboratory of Yellow River Water Environment in Gansu Province, Lanzhou 730070, China
[3] Tianshui Housing and Urban-Rural Development Bureau, Tianshui 741000, China
[4] Beijing General Municipal Engineering Design & Research Institute Co., Ltd., Beijing 100082, China
[*] Correspondence: tssjsjjsk@163.com (H.C.); weibg@mail.lzjtu.cn (B.W.)

**Abstract:** To investigate the quantitative relationship between the volume capture of rainfall and carbon emissions from bioretention facilities, this study introduces the concept of the carbon intensity of volume capture of rainfall. The influence of four key factors—climatic conditions, aquifer height, permeability coefficient, and facility area—was investigated using a residential neighborhood in Tianshui, China, as an example. The results reveal that the carbon intensity value is influenced not only by external environmental changes but also by the inherent attributes of bioretention facilities, such as aquifer height, permeability coefficient, and facility area. The maximum carbon intensity value for the volume capture of rainfall was $-0.0005 \text{ kg CO}_2/\text{m}^3$, while the minimum was $-0.0852 \text{ kg CO}_2/\text{m}^3$, representing a substantial difference of approximately 169 times. Orthogonal experiments identified the facility area as the most significant influencing factor on carbon intensity, with a correlation coefficient of 0.0520. The area of bioretention facilities can be prioritized to meet deployment requirements, taking into account volume capture reduction effects and carbon emissions. For facilities with a high carbon intensity, an emphasis should be placed on enhancing carbon reduction benefits, and various initiatives can be implemented to achieve this goal.

**Keywords:** sponge city; bioretention facility; carbon intensity; carbon reduction benefit



## 1. Research Background

Along with the rapid development of urbanization in China, a myriad of challenges is increasingly hindering the sustainable progress of cities [1]. These challenges encompass issues such as water scarcity, recurrent waterlogging, severe water pollution, and a continuous decline in groundwater levels [2,3]. The coexistence of these problems in modern cities has produced significant threats to the fundamental aspects of residents' lives [4]. China proposed in December 2013 to "build a sponge city with natural accumulation, natural infiltration and natural purification" [5]. A sponge city is inspired by the concept of "Low Impact Development (LID)" [6], a sustainable stormwater management concept that minimizes disasters caused by urban rainfall, prevents water loss, enhances groundwater infiltration and recharge, reduces pollutant discharges, improves urban stormwater storage and utilization, and increases urban resilience [7–9]. With the rapid advancement of ecological civilization construction, prioritizing a green stormwater infrastructure to address urban stormwater management issues has become a crucial strategy for emphasizing ecological priorities and fostering green development in sustainable urban planning [10].

A sponge city mainly includes the construction of LID facilities, such as green roofs, bioretention facilities (BFs), grass-planted swales, permeable pavements and tiles, and infiltration drains [11–13]. In recent years, a substantial number of studies have evaluated the performance of LID facilities individually or in combination with gray infrastructure [14]. Urban flood modeling and flood source tracking methods were used to identify flood source areas and hazard areas, which offered a more nuanced understanding of the optimal

scale for LID facility deployment. Also, various other factors such as different costs [15], different climatic conditions [16,17], etc., were considered at this stage of the study. Davis et al. [18] showed that the quantity of treated water and the effectiveness of water quality are highly dependent on the BF's length, width, slope, type and density of grasses, soil type, quality of runoff, and marsh design. Shao et al. [19] evaluated the potential for green roof deployment by unmanned aerial vehicle, and their results showed that urban-scale LID facilities require different scales of deployment areas to achieve the required scale. Numerous studies have concluded that LID facilities have different stormwater control capabilities depending on their layout (structure, scale) and the environment in which they are located.

Bioretention facilities are stagnant landscaped areas which are designed to attenuate and treat stormwater runoff [20,21]. The bioretention facilities can reduce runoff and peak runoff flows by retaining captured stormwater for a period of time through their filter media [22]. The enhancement of stormwater infiltration by bioretention facilities is primarily achieved by their ability to delay stormwater runoff during peak infiltration periods, and their infiltration capacity is a critical aspect from a hydrological perspective [23]. The soil medium of the bioretention facilities is the most crucial parameter in reducing runoff [24]. Numerous field studies at different sites have confirmed the ability of bioretention systems to reduce stormwater runoff [6,25]. Additionally, the permeable fill media of bioretention basins play a vital role in reducing stormwater pollution [26]. Ou et al. [10] use bioretention facilities as equivalents in an urban sustainability perspective, based on standardized indicators for volume captured per unit area, to simplify and standardize the assessment and planning process for green stormwater infrastructure. Bioretention facilities provide considerable social and environmental advantages through their effective control of stormwater, pollutant reduction, and reasonable investment, establishing them as crucial components of sponge cities [27].

Bioretention facilities, green roofs, and planted swales function as plant carbon sinks, so it is important to continue to conduct in-depth research on the carbon emissions of LID facilities. Her et al. [28] and Cai et al. [29] demonstrated that the carbon reduction pathways of sponge cities, including bioretention facilities, primarily encompass urban runoff volume capture, pollutant emission reduction, air pollution absorption, plant carbon sequestration, urban heat island mitigation, water reuse, etc. Getter et al. [30] observed that green roofs achieve a more effective carbon sink effect. In a related study, Kavehei et al. [31] quantified the carbon emissions and sequestration potential of various LID facilities, including green roofs, rain gardens, bioretention ponds, vegetated buffer strips, and rainwater ponds. Their findings verified that rain gardens possess the highest carbon capacity. Lin et al. [32] developed a comprehensive accounting model based on the Intergovernmental Panel on Climate Change guidelines and life cycle assessment methodology to predict carbon emissions and carbon sinks in sponge cities. The results indicated that sponge cities can achieve a significant amount of carbon sinks during long-term normal operation. It is anticipated that in the second half of the life cycle, the system can attain carbon neutrality. Subsequently, it can be utilized as a carbon emission reduction system to mitigate the greenhouse effect. Su et al. [33] found that sponge cities can reduce carbon emissions from integrated urban drainage systems by an average of 49%, as observed in a comparison with traditional urban drainage systems in Xiamen. Moore et al. [34] conducted a multidimensional comparison of carbon emissions from various LID facilities, encompassing initial emissions, maintenance emissions, and vegetation sequestration over a 30-year period. Their results also showed that rainwater wetlands and bioretention strips exhibited the lowest net carbon emissions per unit area. However, while the calculation methods exhibit some adaptability across different study areas, accurately assessing the magnitude of their carbon emissions and carbon reduction potential remains challenging. Peng et al. [35] calculated the carbon footprint of rain gardens using the life cycle assessment (LCA) methodology, estimated the operational carbon reduction benefits, and concluded that rain gardens have significant potential to mitigate climate change.

The current study focuses on the carbon emission accounting and carbon emission accounting model, exemplified by studies such as Su et al. [33] and Peng et al. [35]. However, due to variations in study area location, size, and climatic conditions, it is challenging to horizontally compare the size of carbon emissions among different study areas. Similarly, within the same study area, the impact of different LID facility layouts on rainfall volume capture varies, and since the volume capture ratio of annual rainfall is the primary goal of sponge city construction, directly comparing the total amount of carbon emissions may lack meaningful significance. Yet, when considering unit quantities, such as intensity, which are often measured in terms of area, there is limited research on the volume capture ratio of annual rainfall (VCRA), a crucial metric for low-impact development facilities. Therefore, we propose the concept of carbon emission intensity based on the volume capture ratio of annual rainfall. This concept can be employed for the comparison of carbon emissions from LID facilities across different study areas and for the evaluation of carbon emissions from various classes of LID facility placement scenarios within the same study area. However, relevant studies on this concept are yet to be seen.

This study innovates by utilizing the storm intensity formula to derive the calculation formula for the total annual runoff control volume. It introduces the concept of carbon emission intensity for the total runoff control volume throughout the entire life cycle and achieves an evaluation of the combined contribution of bioretention facilities to rainfall and carbon emissions.

In this study, considering the effect of sponge city construction, Tianshui City, China, was studied as an example. The variation in carbon emission intensity of volume capture of rainfall from bioretention facilities over the full life cycle under different influencing factors was investigated. The technical route is shown in Figure 1. The results can serve as a reliable foundation for assessing the carbon emissions that come from LID facilities. Additionally, the results can aid in the creation of sponge cities, contributing to their development into more sustainable cities and the implementation of China's "double carbon" strategy.

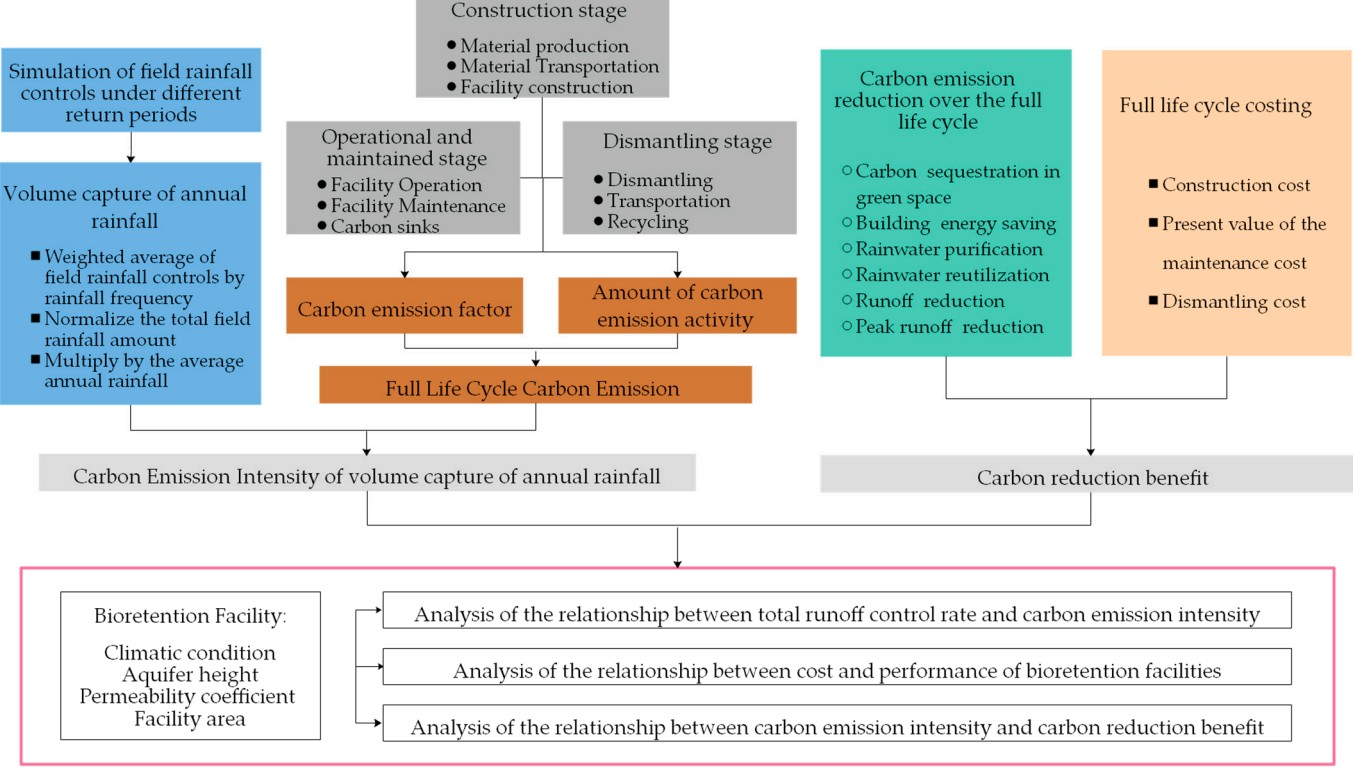

**Figure 1.** Technology roadmap.

## 2. Overview of the Study Area and Research Methods

### 2.1. Overview of the Study Area

The study area is in a newly built residential area in Tianshui, Gansu Province. It is a catchment area within the residential area, and the study simulates the performance of a bioretention facility in this catchment area. The study area is 2930 m$^2$, consisting of built-up areas, green spaces, and roads (including concrete and paved roads). Stormwater pipes and outlets were set up based on the elevation and pipe layout of the study area. The natural drainage direction is from northwest to southeast, as shown in Figure 2.

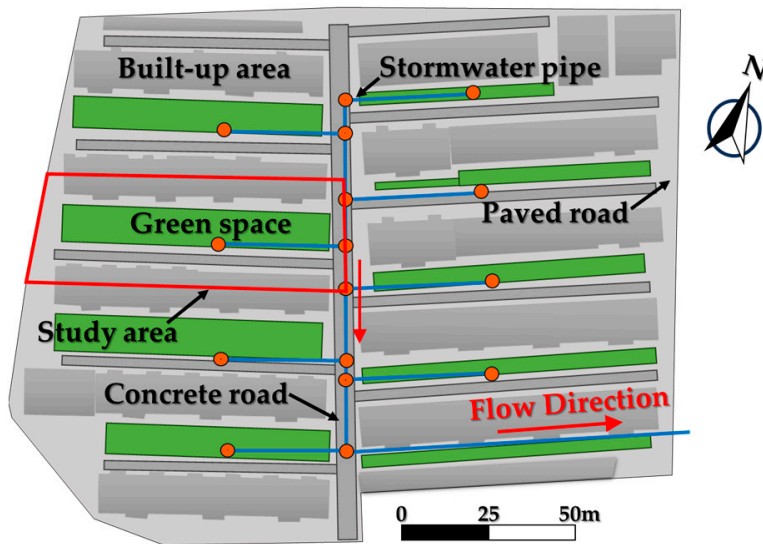

**Figure 2.** Schematic map of the study area.

### 2.2. Model Construction and Research Methods

2.2.1. Design Rainfall Intensity

The rainfall intensity in Tianshui was calculated by Equations (1) and (2):

$$q = \frac{712.900(1 + 1.90 \times \lg(p))}{(t + 8.711)^{0.742}}(p < 20 \text{ a}) \tag{1}$$

$$q = \frac{1336.703(1 + 1.96 \times \lg(p))}{\left(t + 12.940\right)^{0.840}}(p \geq 20 \text{ a}) \tag{2}$$

where $q$ represents the rainfall intensity in L/(s·ha), $p$ represents the design rainfall return period in a, and $t$ represents the rainfall duration in min.

Using the Chicago Rainfall Type Generation Software (1.0), a rainfall intensity curve with a peak coefficient of 0.398 was generated based on Equation (1) for design rainfall return periods of 0.5 a, 1 a, 2 a, 3 a, 5 a, and 10 a, with a rainfall duration of 120 min (Figure 3). Similarly, a long-duration rainfall curve with a rainfall duration of 1440 min was generated based on Equation (2) for design rainfall return periods of 20 a, 30 a, and 50 a, as shown in Figure 4.

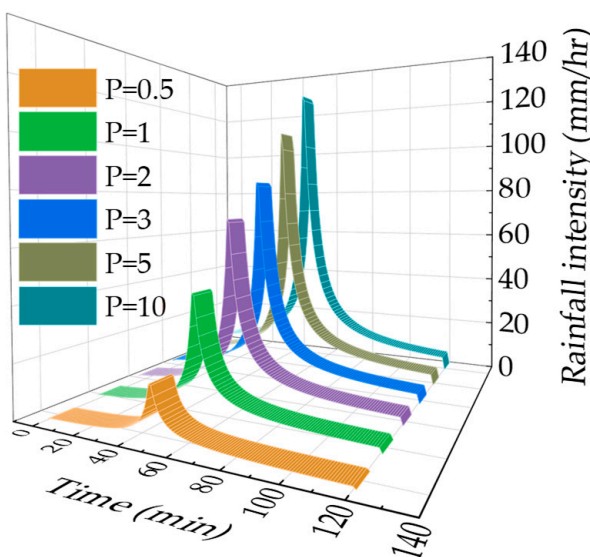

**Figure 3.** 120 min short-duration rainfall intensity.

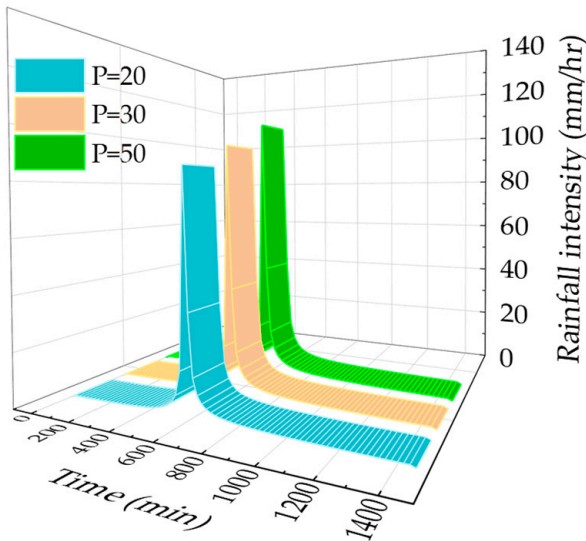

**Figure 4.** 1440 min long-duration rainfall intensity.

### 2.2.2. InfoWorks ICM Model Construction and Parameter Calibration

The InfoWorks ICM (2023.1) was selected for simulation. This software is a comprehensive urban watershed drainage model that can simulate the entire urban rainwater cycle system. It can simulate and evaluate the performance of various rainwater storage and utilization facilities, providing technical support for preventing urban waterlogging and controlling and utilizing rainwater resources [36]. The rainfall–runoff model was constructed based on the topography and design drawings of the residential area, and it was generalized into 8 sub-catchment areas, 16 nodes, and 16 pipes. The study area is one of the sub-catchment areas, as shown in Figure 1.

The runoff coefficient method is the parameter rate determination method of the urban stormwater model selected in this paper [37]. According to the "Standard for Design of Outdoor Wastewater Engineering (GB 50014-2021)" and considering the characteristics of the site's underlying surface and soil, the runoff coefficients for roof surfaces, concrete roads, paved roads, and green spaces were set as 0.90, 0.85, 0.60, and 0.15, respectively. The comprehensive runoff coefficient for the site was calculated through area weighting, resulting in a value of 0.66. After establishing the model for the site, a simulation verification

was conducted using a rainfall of 26.58 mm with a two-year return period under normal rainfall conditions. The resulting runoff coefficient for the site was also 0.66, confirming the rationality of the model parameters that were set in this study.

2.2.3. Experimental Design

The basic parameters of the BF were determined based on the relevant specifications for sponge city construction [38] and the existing literature [28,39]. The selected bioretention facility is an anti-seepage-shaped BF, referred to as the BF. The influencing factors and scope of bioretention facilities are defined by domestic regulations and relevant international norms. These guidelines emphasize the significance of considering aquifer height, permeability coefficient, and facility area as crucial factors in the construction process [40,41]. Additionally, within the context of climate change, the study of the impact of climate conditions on bioretention facilities has been introduced. The climate conditions varied within ±20% of the rainfall amount, calculated using the local rainfall intensity formula with a step size of 10%. The ranges for the aquifer height and permeability coefficient were set at 100 mm to 300 mm (with a step size of 50 mm) and 50 mm/h to 150 mm/h (with a step size of 25 mm/h), respectively. The facility areas are 45.5 m², 72.8 m², 91 m², 136.5 m², 182 m², and 273 m², which corresponds to 5%, 8%, 10%, 15%, 20%, and 30% of the proportion of the facility area to the green area. The corresponding values for each influencing factor are shown in Table 1.

**Table 1.** Values of influencing factors.

| Influencing Factor | Value | | | | | |
|---|---|---|---|---|---|---|
| Climatic condition [1] (CC) | C−20 | C−10 | C | C+10 | C+20 | |
| Aquifer height (AH) (mm) | 100 | 150 | 200 | 250 | 300 | |
| Permeability coefficient (PC) (mm/h) | 10 | 25 | 50 | 100 | 200 | |
| The facility area (FA) (m²) | 45.5 | 72.8 | 91 | 136.5 | 182 | 273 |

Note: [1] C−20, C−10, C, C+10, and C+20 represent rainfall amounts corresponding to 80%, 90%, 100%, 110%, and 120% of the rainfall calculated using the local rainfall intensity formula, respectively.

In order to determine the value of the carbon intensity and carbon reduction effect under different levels of single influencing factors, 19 simulation scenarios under different influencing factors were established through the method of single-factor experiment (scenario numbers are 1–18 and 36). Afterwards, 24 scenarios were established through orthogonal experiments, which were scenarios 19–42, to determine the strength of the correlation between the individual influencing factors on the carbon intensity and carbon reduction benefits. The details of each scenario are shown in Table 2.

**Table 2.** Simulation scenarios.

| Scenario Number | Factors | | | | Scenario Number | Factors | | | |
|---|---|---|---|---|---|---|---|---|---|
| | CC | AH (mm) | PC (mm/h) | FA (m²) | | CC | AH (mm) | PC (mm/h) | FA (m²) |
| 1 | 0.8 | 200 | 50 | 91 | 22 | 0.8 | 300 | 100 | 91 |
| 2 | 0.9 | 200 | 50 | 91 | 23 | 0.9 | 100 | 200 | 136.5 |
| 3 | 1.1 | 200 | 50 | 91 | 24 | 0.9 | 150 | 25 | 72.8 |
| 4 | 1.2 | 200 | 50 | 91 | 25 | 0.9 | 200 | 100 | 182 |
| 5 | 1 | 100 | 50 | 91 | 26 | 0.9 | 250 | 10 | 91 |
| 6 | 1 | 150 | 50 | 91 | 27 | 0.9 | 300 | 50 | 45.5 |
| 7 | 1 | 250 | 50 | 91 | 28 | 1 | 100 | 100 | 72.8 |
| 8 | 1 | 300 | 50 | 91 | 29 | 1 | 150 | 10 | 182 |
| 9 | 1 | 200 | 10 | 91 | 30 | 1 | 200 | 50 | 91 |
| 10 | 1 | 200 | 25 | 91 | 31 | 1 | 250 | 200 | 45.5 |

**Table 2.** *Cont.*

| Scenario Number | Factors | | | | Scenario Number | Factors | | | |
|---|---|---|---|---|---|---|---|---|---|
| | CC | AH (mm) | PC (mm/h) | FA (m²) | | CC | AH (mm) | PC (mm/h) | FA (m²) |
| 11 | 1 | 200 | 100 | 91 | 32 | 1 | 300 | 25 | 136.5 |
| 12 | 1 | 200 | 200 | 91 | 33 | 1.1 | 100 | 50 | 182 |
| 13 | 1 | 200 | 50 | 45.5 | 34 | 1.1 | 150 | 200 | 91 |
| 14 | 1 | 200 | 50 | 72.8 | 35 | 1.1 | 200 | 25 | 45.5 |
| 15 | 1 | 200 | 50 | 136.5 | 36 | 1.1 | 250 | 100 | 136.5 |
| 16 | 1 | 200 | 50 | 182 | 37 | 1.1 | 300 | 10 | 72.8 |
| 17 | 1 | 200 | 50 | 273 | 38 | 1.2 | 100 | 25 | 91 |
| 18 | 0.8 | 100 | 10 | 45.5 | 39 | 1.2 | 150 | 100 | 45.5 |
| 19 | 0.8 | 150 | 50 | 136.5 | 40 | 1.2 | 200 | 10 | 136.5 |
| 20 | 0.8 | 200 | 200 | 72.8 | 41 | 1.2 | 250 | 50 | 72.8 |
| 21 | 0.8 | 250 | 25 | 182 | 42 | 1.2 | 300 | 200 | 182 |

2.2.4. Calculation Method for Rainwater Control Performance

The rainwater control performance is represented by the volume capture ratio of annual rainfall (*VCRA*) and the volume capture of annual rainfall (*VCA*).

The *VCRA* is calculated as the weighted average of the difference between the rainfall and runoff volumes, as shown in Equation (3). The *VCA* is calculated as the *VCRA* divided by the annual rainfall volume, as shown in Equation (4).

$$VCRA = \frac{\sum_{j=1}^{9} \frac{V_{\text{rain},j} - V_{\text{runoff},j}}{P_j}}{A \sum_{j=1}^{9} \frac{H_j}{P_j}} \times 100\%$$ (3)

$$VCA = VCRA \times H_a \times A$$ (4)

where *VCRA* represents the volume capture ratio of annual rainfall in %, $P_j$ represents the return period ($j$ = 1–9 represents return periods of 0.5, 1, 2, 3, 5, 10, 20, 30, and 50 years), $V_{\text{rain},j}$ represents the rainfall volume for a return period of $P_j$ in m³, $V_{\text{runoff},j}$ represents the runoff volume generated when the LID facility is in place for a return period of $P_j$ (i.e., the overflow from the BF) in m³, $H_a$ is the annual average rainfall volume in the study area in mm (set as 505 mm), $H_j$ indicates the rainfall intensity for a return period of $P_j$ in mm, *VCA* represents the volume capture of annual rainfall in m³, and $A$ indicates the size of the study area in m² (set as 2930 m²). The value of $V_{\text{runoff},j}$ was calculated using the InfoWorks ICM model.

2.2.5. Full Life Cycle Costing Calculation Method

This study only analyzed the full life cycle costing of LID facilities, including construction, operation and maintenance, and removal costs, without considering other required costs (Figure 5). The life cycle of the BF is set to 30 years in this paper. The calculation formula refers to existing research results [42–46], such as Equations (5)–(8):

$$FLCC = C_{\text{Capital}} + \sum_{t=1}^{n_k} PV_{O\&M_{k,t}} + C_{\text{remove}}$$ (5)

$$PV_{O\&M_t} = \frac{FV_{O\&M_t}}{(1+d)^t}, \ \forall t$$ (6)

$$FV_{O\&M_t} = C_{\text{Capital}} \times p \times (1+r)^t, \ \forall t$$ (7)

$$C_{\text{dismantle}} = y \times C_{\text{Capital}} - S$$ (8)

where *FLCC* represents the full life cycle costing of the BF, in CNY. $n_k$ refers to the life cycle of the LID facility in years, taking $n_k$ = 30. $C_{\text{Capital}}$ is the construction cost of the

facility in CNY, and the unit price is referenced from relevant research [47] and engineering examples in the northwest region of China. $PV_{O\&M_t}$ represents the present value of the maintenance cost of the LID facility in year $t$ in CNY. $FV_{O\&M_t}$ represents the future value of the maintenance cost of the LID facility in year $t$ in CNY. $d$ is the discount rate in %, taking 8% as the current social discount rate. $r$ is the average inflation rate in %, taking 3%. $p$ is the proportion of annual operation and maintenance costs to construction costs in %, taking $p = 8$. $C_{dismantle}$ is the cost of facility dismantle in CNY. $y$ is the proportion of removal cost to construction cost in %, taking 10%. $S$ is the salvage value, taking $S = 0$.

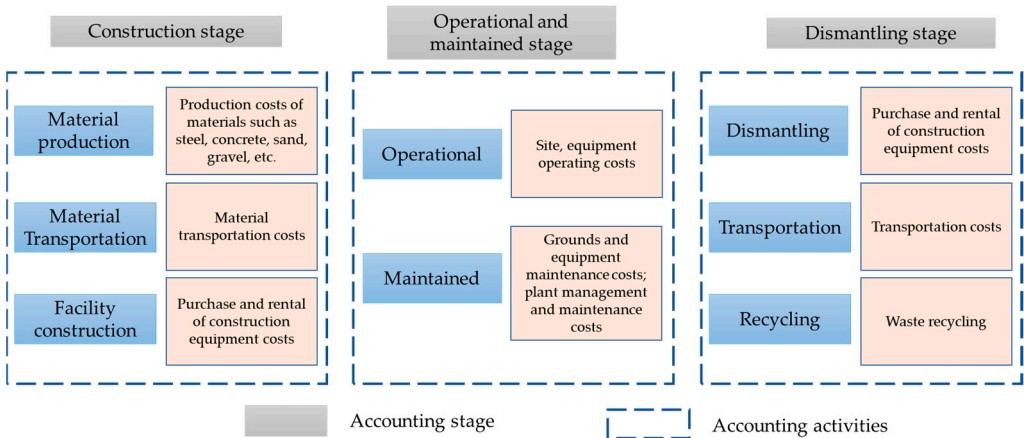

**Figure 5.** Full life cycle costing accounting boundary and activities.

### 2.2.6. Full Life Cycle Carbon Emission Accounting Method

The full life cycle carbon emission accounting of LID facilities includes carbon emissions during the construction, operation and maintenance, and dismantling stages of the LID facility (Figure 6), calculated using the carbon emission factor method. The life cycle carbon emissions are calculated using Equations (9)–(12).

$$CE = \frac{30}{n_k}(CE_{CS} + CE_{OM} + CE_{DS}) \tag{9}$$

$$CE_{CS} = AD \times EF_{CS} \tag{10}$$

$$CE_{OM} = VCA \times EF_O + AD \times EF_M - AD \times EF_H \tag{11}$$

$$CE_{DS} = AD \times EF_{DS} \tag{12}$$

where $CE$ represents the full life cycle carbon emissions of the BF in kg $CO_2$. $CE_{CS}$ and $CE_{DS}$ represent the carbon emissions during the construction stage and dismantling stage of the LID facility in kg $CO_2$. $CE_{OM}$ represents the carbon emission during the operation and maintenance stage of the LID facility's full life cycle in kg $CO_2$. $AD$ represents the area of the BF in $m^2$. $EF_{CS}$ and $EF_{DS}$ represent the carbon emission factors during the construction and dismantling stages of the LID facility in kg $CO_2/m^2$. $EF_O$ represents the operational carbon emission factor of the LID facility during the full life cycle in kg $CO_2/m^3$. $EF_M$ represents the maintained carbon emission factor of the LID facility during the full life cycle in kg $CO_2/m^2$. And $EF_H$ represents the carbon sink factor of the carbon sequestration for the green space of the LID facility full life cycle in kg $CO_2/m^2$. The values of some constants in the formulas are shown in Table 3.

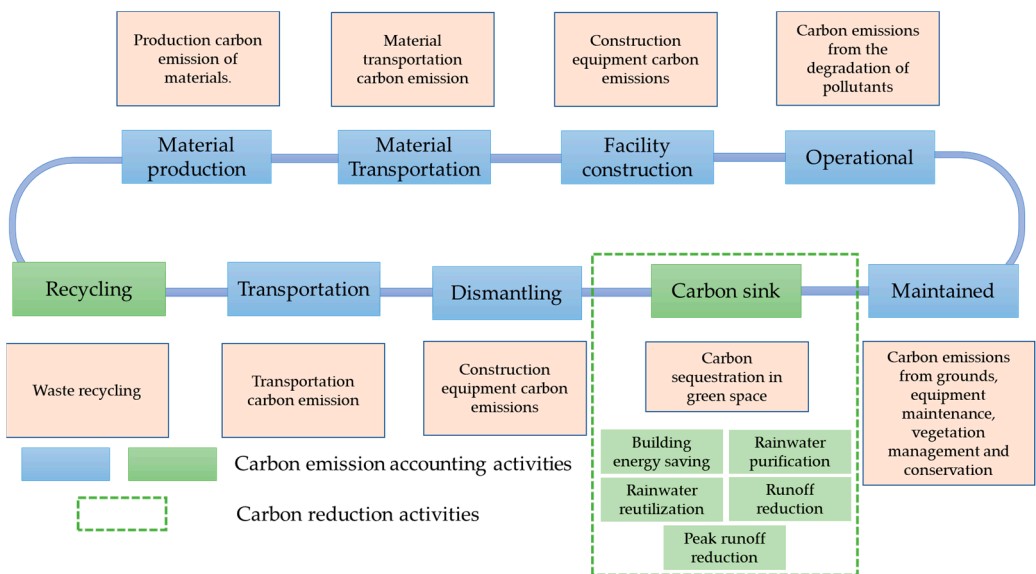

**Figure 6.** Full life cycle carbon emission accounting boundary and activities.

**Table 3.** Constant values.

| Constant | $EF_{CS}$ (kg $CO_2$/m²) | $EF_{DS}$ (kg $CO_2$/m²) | $EF_O$ (kg $CO_2$/m³) | $EF_M$ (kg $CO_2$/m²) | $EF_H$ (kg $CO_2$/m²) | $EF_{HL}$ (kg $CO_2$/m³) |
|---|---|---|---|---|---|---|
| Value [1] | 44.2523 | 5.6710 | 3.7890 | 5.1300 | 66.9000 | 44.6160 |

Note: [1] The constant values concerning the relevant carbon emission factors are taken from studies such as Li [48], Ma [49], and Li [50] by accounting.

### 2.2.7. Concept and Calculation Method of Carbon Intensity of *VCA* Based on Full Life Cycle

Based on the LCA theory, the carbon emission intensity of *VCA*, defined as the ratio of carbon emissions of the facility over the runoff control volume throughout its life cycle, is proposed. It is used to quantitatively evaluate the relationship between the rainwater control capacity and carbon emissions of LID facilities. It is calculated according to Equations (13) and (14). A smaller value of this indicator indicates fewer carbon dioxide emissions per unit of controlled runoff, leading to better environmental benefits.

$$E = \frac{CE}{CV_{30}} \tag{13}$$

$$CV_{30} = 30 \times VCA \tag{14}$$

where *E* represents the carbon emission intensity of the runoff control volume in kg $CO_2$/m³ of volume capture of annual rainfall. $CV_{30}$ is the *VCA* of full life cycle in m³. Other symbols have the same meanings as before.

### 2.2.8. Concept and Calculation Method of Carbon Reduction Benefit Based on Full Life Cycle

The concept of a carbon reduction benefit is proposed based on the cost-effectiveness [51] calculation method and the LCA theory. It refers to the ratio of carbon reduction of the LID facility over its cost throughout its life cycle. It is used to evaluate the relationship between the cost and carbon emission reduction of the LID facility and is calculated according to Equations (15) and (16). A higher value of carbon reduction benefit indicates higher economic effectiveness and value of carbon emission reduction in the construction of LID facilities.

$$B = \frac{CER}{LCC} \tag{15}$$

$$CER = AD \times EF_H + VCA \times EF_{HL} \tag{16}$$

where $B$ represents the carbon reduction benefit in kg $CO_2$/CNY, $CER$ represents the carbon emission reduction over the full life cycle of the LID facility, and the value is equal to the carbon emission reduction generated by activities such as carbon sequestration in green spaces, rainwater utilization, runoff reduction, and rainwater purification in kg $CO_2$, and $EF_{HL}$ indicates the combined carbon emission factor for rainwater utilization and rainwater purification over the full life cycle of the LID facility in kg $CO_2$/m$^3$. Other symbols have the same meanings as before. The values of some constants in the formulas are shown in Table 3.

## 3. Results and Discussion

### 3.1. Compositional Analysis of Carbon Emission from BF in Full Life Cycle

The proportion of carbon emissions from stages or activities of the full life cycle of the BF in scenario 32 is shown in Figure 7. According to Equations (9)–(12), the carbon emission in the full life cycle can be calculated as follows: 5895.33 kg $CO_2$ for the construction stage, 921.56 kg $CO_2$ and 700.25 kg $CO_2$ for the operation activity and maintenance activity in the operation and maintenance stage, and 80.54 kg $CO_2$ for the dismantling phase. In addition, we calculated −9131.85 kg $CO_2$ for the carbon sink of the full life cycle. The pure carbon emission (sum of carbon emissions and sinks) during the operation and maintenance stage is −7524.16 kg $CO_2$, and the pure carbon emission during the full life cycle is −1458.10 kg $CO_2$. The carbon emission accounting results of the full life cycle of bioretention facilities in this study differ from some existing research results [32,52] because of the inconsistency in the definition of carbon sinks in the boundary of full life cycle accounting. The related studies' results include carbon sequestration in green spaces, rainwater purification, and rainwater utilization of the operation and maintenance phase of the bioretention facilities as carbon sinks in the accounting system. In this paper, referring to the relevant literature [53], only the activities that produce resources or energy in the production process and can be delivered to the outside are defined as carbon sinks, and the actions that meet this condition are the greenfield carbon sequestration in the operation and maintenance stage of bioretention facilities. Other activities in the operation and maintenance phase of bioretention facilities, such as rainwater purification and utilization, do not satisfy this study's definition of carbon sinks. Together with the sink activities, they are defined as carbon emission reductions, which is outside the boundary of the full life cycle accounting and is discussed separately in Section 3.3.

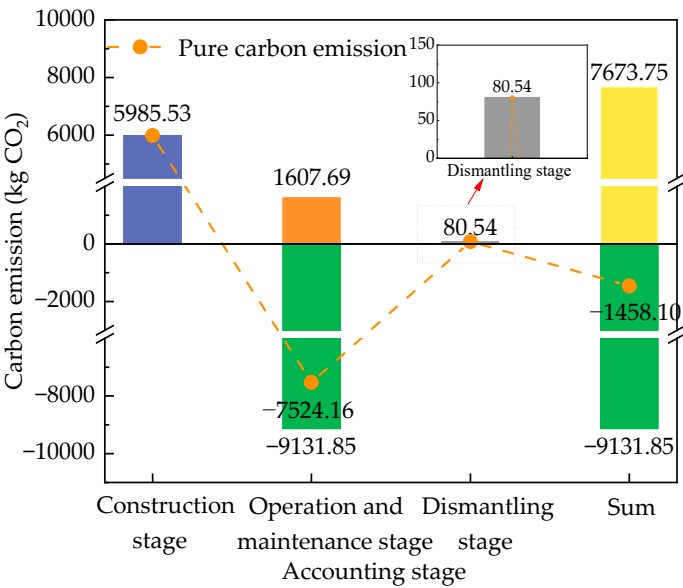

Note: Negative values represent carbon sinks.

**Figure 7.** Schematic diagram of the composition of carbon emissions of BF over the full life cycle.

### 3.2. *Analysis of Carbon Emission Intensity under Different Influencing Factors*

#### 3.2.1. Climate Condition

As shown in Figure 8a, climate change affects rainwater control performance: as the intensity of storms increases, the *VCA* of the facility increases, which shows that the BF has a stormwater control capacity for increasing rainfall. The life cycle carbon emissions under different climatic conditions ranged from $-874.1175$ kg $CO_2$ to $-657.3433$ kg $CO_2$. The carbon intensity ranged from $-0.0306$ kg $CO_2/m^3$ to $-0.0176$ kg $CO_2/m^3$, with a changing trend that is consistent with the life cycle carbon emissions and a variation range of 74%. The full life cycle carbon emissions gradually increased with the increase in rainfall because of the constant parameters of the BF. Carbon emissions and carbon sinks stay unaltered during the construction and demolition stages. However, the change in climatic conditions led to a shift in rainfall, and the organic substance in the corresponding runoff was treated by the BF, which increased the direct carbon emissions during the operation and maintenance stage, increasing life cycle carbon emissions.

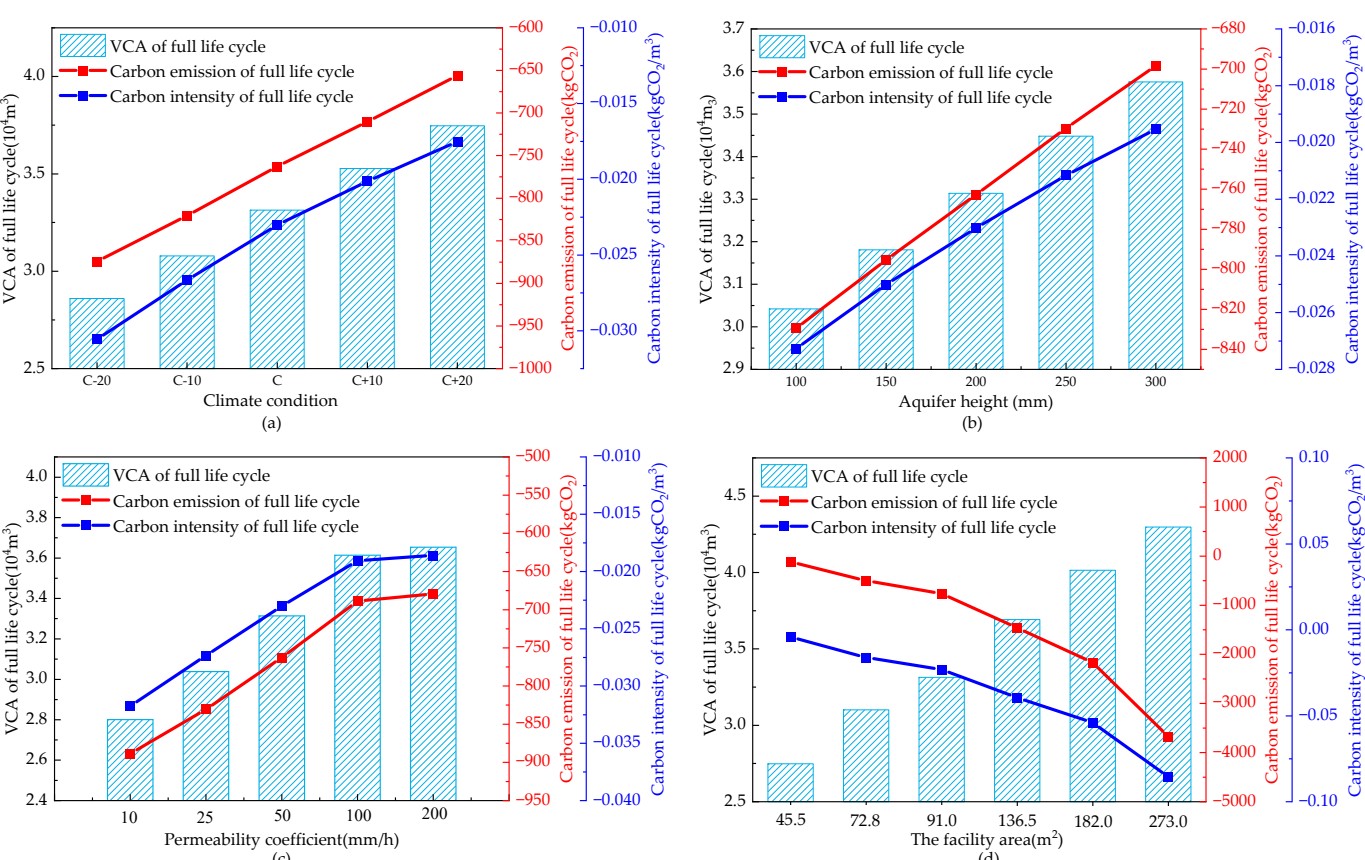

**Figure 8.** Impact of (**a**) climatic conditions, (**b**) aquifer height, (**c**) permeability coefficient, and (**d**) the facility area on carbon emissions and carbon intensity.

#### 3.2.2. Aquifer Height

As shown in Figure 8b, the larger the aquifer height is, the stronger the ability of rainwater capture is, which is mainly reflected in the increase in *VCA*. According to the increasing height of the aquifer, *VCA* showed an increasing trend, and the higher the height of the aquifer was, the more significant the benefit of the facility for runoff control was, which is similar to the results of the study by Tu et al. [54]. As rainwater runoff flows into the BF, one part infiltrates through the soil layer, and the other part is stored in the aquifer. The higher the height of the aquifer is, the more stormwater runoff can be held in the facility, and the greater the storage capacity for stormwater runoff is. The life cycle carbon emissions of the BF increased from $-827.2060$ kg $CO_2$ at an aquifer height of 100 mm to

$-695.6845$ kg $CO_2$ at an aquifer height of 300 mm. The carbon emission intensity increased monotonically with an increase in aquifer height, ranging from $-0.0271$ kg $CO_2/m^3$ to $-0.0194$ kg $CO_2/m^3$, with a variation range of 40%. The increased aquifer height allows the BF to retain more rainwater during the operation and maintenance stage. Therefore, the amount of rainwater purification is high, and the direct carbon emissions during the operation phase are correspondingly high.

### 3.2.3. Permeability Coefficient

As shown in Figure 8c, the PC of the BF follows the same trend as the stormwater control performance. The PC resulting from 10–100 mm/h changes in the stormwater control effect have the greatest impact. With the increase in the infiltration coefficient, the impact of bioretention facilities on the stormwater volume capture effect gradually decreases. This is because during rainfall, as runoff rainwater flows into the soil void, the soil infiltration coefficient gradually decreases and eventually reaches a stabilized value. Higher initial PCs can slow this process, and the runoff control effect improves. This is similar to the findings of Pan et al. [55]. The life cycle carbon emissions of the BF ranged from $-886.6840$ kg $CO_2$ to $-676.5319$ kg $CO_2$ under soil PCs of 50 mm/h to 150 mm/h. The carbon emission intensity values increased monotonically with an increase in soil PC, ranging from $-0.0316$ kg $CO_2/m^3$ to $-0.0185$ kg $CO_2/m^3$. The carbon emission intensity at a 200 mm/h PC increased by 71% compared to that at a 50 mm/h permeability coefficient because changes in soil permeability affected the stormwater management capacity of the BF, as demonstrated by Haaland et al. [56]. This, in turn, affected the amount of stormwater that was purified by the BF. And it could also affect carbon emissions during the operation and maintenance stage. In the life cycle assessment boundary of this study, changes in soil permeability had no effect on the variance in carbon emission intensity during the construction and demolition stages. Therefore, an increase in soil PC led to a rise in life cycle carbon intensity.

### 3.2.4. The Facility Area

The FA will directly affect the facility's water intake, which is one of the leading design parameters of the BF. As seen in Figure 8d, the greater its value is the better the stormwater control performance eventually becomes. This is because the change in the layout area does not affect the rate of soil infiltration, and overall, with the increase in the layout area ratio, the inlet flow decreases, and the facility's ability to regulate runoff tends to increase. The full life cycle carbon emissions increased monotonically with an increase in the ratio of the BF area to the green space area. When the BF accounted for 30% of the green space area, the life cycle carbon emission was $-0.0852$ kg $CO_2/m^3$. The carbon emission intensity changes with a runoff reduction volume from 5% to 30% follow a diverse trend in terms of the life cycle carbon emissions, which decrease from $-0.0040$ kg $CO_2/m^3$ to $-0.0852$ kg $CO_2/m^3$ with a variation range of 95%. This phenomenon may be explained via the life cycle theory. The increase in the facility area lead to an increase in the carbon emission intensity during the construction, operation and maintenance, and dismantling stages. At the same time, the expansion of the bioretention facility area results in an increase in the amount of greenery and impermeable surfaces, which strengthens the carbon sink effect [57]. The increased carbon sink throughout the life cycle can offset part of the increased carbon emission intensity in the construction, operation and maintenance, and demolition phases, slowing down the trend of increasing full life carbon emissions. Therefore, the larger the area of bioretention facilities is, the more significant the carbon sequestration effect is and the lower the carbon emissions are.

### 3.3. Analysis of Carbon Reduction Benefits of Different Influencing Factors

#### 3.3.1. Climate Condition

Figure 9a shows that during the change from climate condition C−20 to C+20, the carbon emission reduction throughout the life cycle of the BF ranged from $4.2899 \times 10^4$ kg $CO_2$

to $5.4251 \times 10^4$ kg $CO_2$. The carbon reduction benefit increased from 1.5363 kg $CO_2$/CNY in the C−20 period to 1.9429 kg $CO_2$/CNY in the C+20 period, with an increase of 30.6%. The influence of changes in the climate condition on different carbon emission reduction measures is observed, as the carbon emission reduction from green space carbon seques­tration remains constant due to the constant area of deployment. The carbon emission reduction from rainwater utilization is the highest, the rainwater purification carbon emis­sion reduction is the second highest, and both increase with the increase in rainfall. These findings suggested that variations in external CSs provided an increase in the BF's brilliant carbon reduction capability. This is because the facility's structure, including storage, infiltration, and drainage layers, can preserve rainwater and indirectly contribute to carbon reduction through factors such as rainwater utilization. This observation is consistent with the findings of Cai [29]. Taking C+10 as an example, under this climatic condition, the BF can achieve a carbon reduction benefit of 1.8422 kg $CO_2$ per unit cost, indicating good economic value.

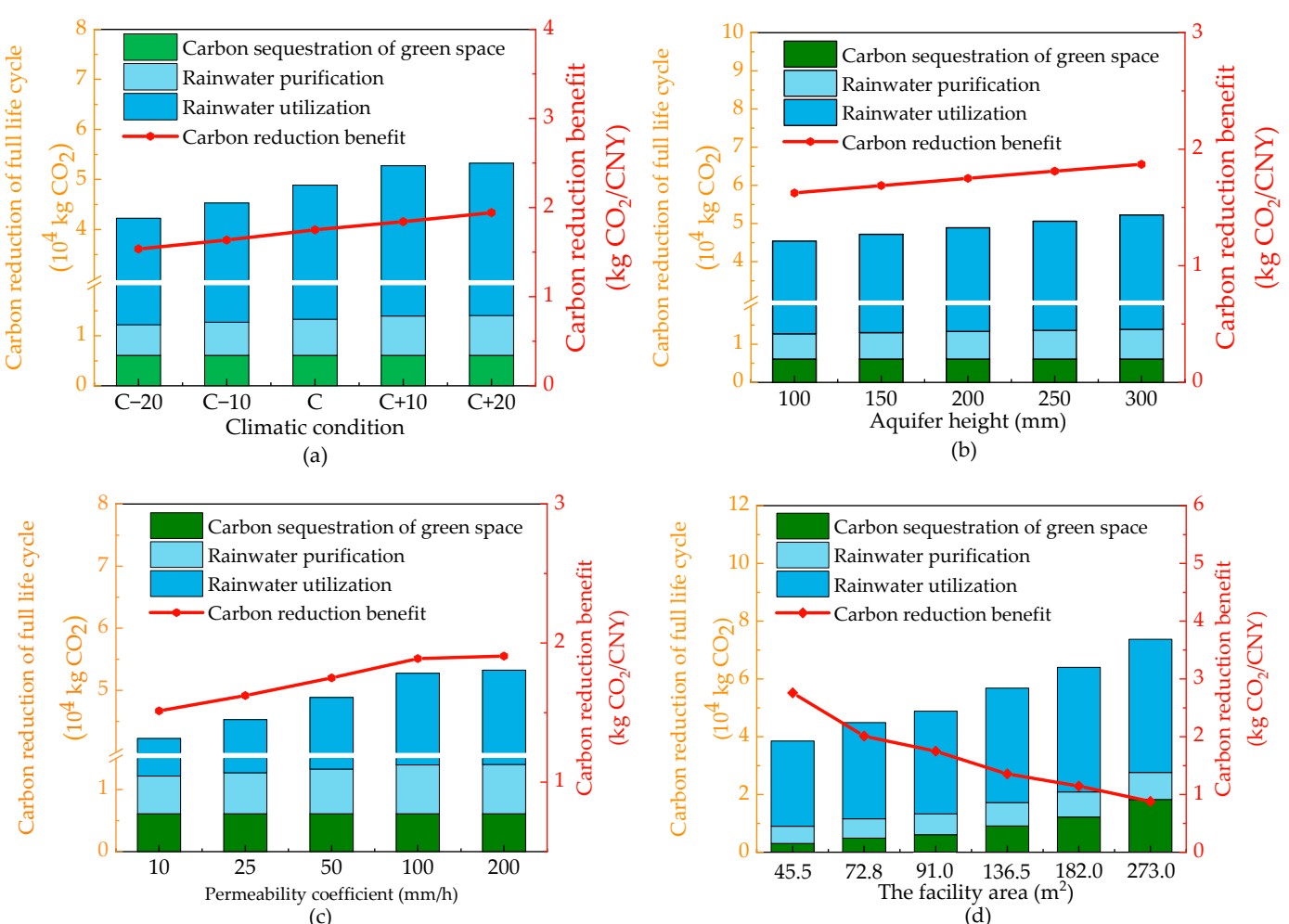

**Figure 9.** Effects of (**a**) climatic conditions, (**b**) aquifer height, (**c**) permeability coefficient, and (**d**) the facility area on the carbon reduction benefit.

### 3.3.2. Aquifer Height

Figure 9b shows that the carbon reduction throughout the full life cycle of the BF grad­ually increased with the increase in aquifer height, ranging from $4.5356 \times 10^4$ kg $CO_2$ to $5.2244 \times 10^4$ kg $CO_2$. The carbon reduction benefit increased gradually as the aquifer height increased from 1.6243 kg $CO_2$/CNY for an aquifer height of 100 mm to 1.8710 kg $CO_2$/CNY for an aquifer height of 300 mm. When compared to a storage layer height of 100 mm,

each increase of 50 mm can result in an increase of approximately 5.0% to 14.4% in carbon reduction benefits, because the BF can retain more rainwater and contribute to increased carbon reduction through rainwater purification and utilization, resulting in a higher carbon reduction potential. Increasing the height of the aquifer will significantly reduce the use of tap water and conserve rainwater resources [29]. This result was also demonstrated by Moore et al. [58].

### 3.3.3. Permeability Coefficient

Figure 9c shows that as the PC increased, the carbon emission reduction throughout the full life cycle of the BF increased, and the carbon reduction benefit increased from 1.5128 kg $CO_2$/CNY to 1.9069 kg $CO_2$/CNY, with an increase of 21%. With the increase in the PC of the BF, permeability pairs affected the carbon reduction capacity. The infiltration rate of rainwater increased, and the rainwater infiltrated into the drainage layer and was discharged through blind pipes, which can reduce the total runoff and peak flow, resulting in increased carbon sequestration. Plants and the composition of the soil medium are significant factors influencing infiltration coefficients and are critical to the ability of bioretention areas to remove pollutants from stormwater and retain stormwater. Therefore, it can be inferred that carbon reduction activities are equally important for both stormwater purification and stormwater utilization in these areas [59].

### 3.3.4. The Facility Area

Figure 9d shows that as the FA increased, the carbon emission reduction throughout the life cycle of the BF gradually increased from $3.8515 \times 10^4$ kg $CO_2$ to $7.3734 \times 10^4$ kg $CO_2$. This is due to an increase in the area of bioretention facilities, an increase in the carbon sink capacity of the greenfield carbon sink, and an increase in the carbon emission reduction capacity of rainwater purification and rainwater utilization, all of which contribute to a gradual increase in full-life carbon emission reductions. The carbon reduction benefits for a layout area ratio of 5% to 30% are 2.7586 kg $CO^2$/CNY, 2.0103 kg $CO_2$/CNY, 1.7499 kg $CO_2$/CNY 1.3560 kg $CO_2$/CNY, 1.1460 kg $CO_2$/CNY, and 0.8802 kg $CO_2$/CNY, and the carbon reduction benefit gradually decreases with the increase in the deployment area, with a decrease of 213%. Compared to the growth in carbon reduction across the life cycle, the increase in the FA resulted in a more noticeable increase in the overall cost of the life cycle. Therefore, the layout area's ratio has a substantial impact on the overall life cycle cost, which is consistent with the conclusions drawn by Wang et al. [60].

### 3.4. Orthogonal Experiment

#### 3.4.1. Carbon Intensity

Figure 10 illustrates the outcomes in terms of carbon emission of the volume capture and carbon intensity throughout the full life cycle for various scenarios related to the bioretention facility. These scenarios correspond to numbers 19–42, as outlined in Table 2.

In the 25 sets of scenarios, the reduction in the full life cycle runoff control values varied between 16,582.2064 m$^3$ and 50,697.7611 m$^3$. Additionally, the full life cycle carbon emissions exhibited a range from $-2357.9866$ kg $CO_2$ to $-14.2365$ kg $CO_2$, while the carbon intensity of the volume capture of rainfall ranged from $-0.0728$ kg$CO_2$/m$^3$ to $-0.0005$ kg$CO_2$/m$^3$. To validate the impacts of various influencing factors on the bioretention facility's performance, orthogonal experiments were conducted using a level difference analysis, and the results are shown in Table 4.

According to the results of the extreme difference analysis for carbon intensity in Table 4, the analysis reveals that the coefficients (R) for different factors—climatic condition, aquifer height, permeability coefficient, and the facility area—are 0.0150, 0.0090, 0.0150, and 0.0520, respectively. A higher R value in the extreme difference indicates a greater impact of the corresponding factor.

The influence of the four factors on the volume capture and carbon emission intensity is analyzed in the following order: the facility area, permeability coefficient, climate condition,

and aquifer height. Therefore, in the design of bioretention facilities, priority can be given to the area ratio of bioretention facilities, considering its significant impact on the volume capture and carbon emissions. This study also found that, in comparison to other design characteristics, the area occupied by bioretention facilities has a greater impact on the intensity of carbon emissions.

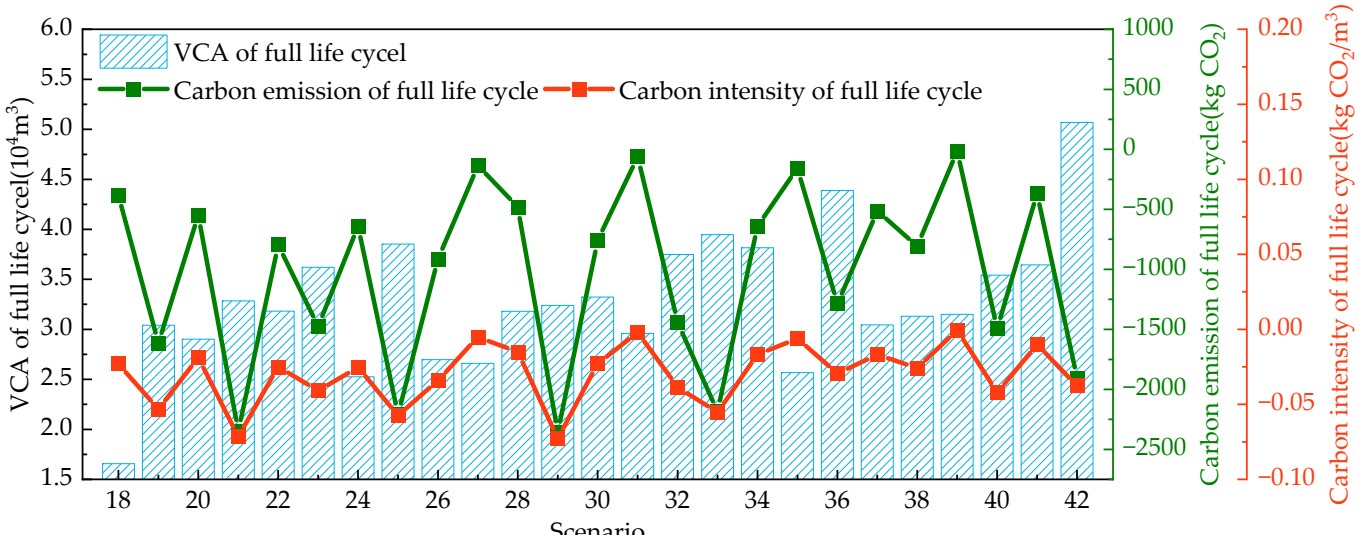

**Figure 10.** Results of scenarios 19–42 on carbon intensity of volume capture of rainfall.

**Table 4.** Orthogonal experiment results for carbon intensity.

|  | CC | AH | PC | FA |
|---|---|---|---|---|
| K1 | −0.1513 | −0.1891 | −0.1813 | −0.0379 |
| K2 | −0.1637 | −0.1488 | −0.1565 | −0.0730 |
| K3 | −0.1603 | −0.1885 | −0.1476 | −0.1101 |
| K4 | −0.1489 | −0.1240 | −0.1611 | −0.2351 |
| K5 | −0.1639 | −0.1377 | −0.1415 | −0.3320 |
| R | 0.0150 | 0.0090 | 0.0150 | 0.0520 |

### 3.4.2. Carbon Reduction Benefit

The full life cycle carbon reduction and carbon reduction benefit of different experimental schemes of bioretention facilities are shown in Figure 11. The full life cycle carbon reduction is $2.4385 \times 10^4$–$7.7423 \times 10^4$ m³, and the full life cycle carbon reduction benefits are 0.8802–3.1223 kg $CO_2$/CNY. To validate the impact of different influencing factors on the bioretention facilities, an analysis of the orthogonal experiment was carried out to perform the level difference analysis, and the results are shown in Table 5.

According to the results of the extreme difference analysis for carbon emission intensity in Table 5, it is observed that the R of the different factors climatic condition, aquifer height, permeability coefficient, and the facility area are 0.4690, 0.2700, 0.4160, and 1.5020, respectively. Specifically, the order is the facility area, permeability coefficient, climate condition, and aquifer height. Consequently, in the design of bioretention facilities, expanding the installation area complements efforts to enhance both the total runoff control and carbon emission reduction.

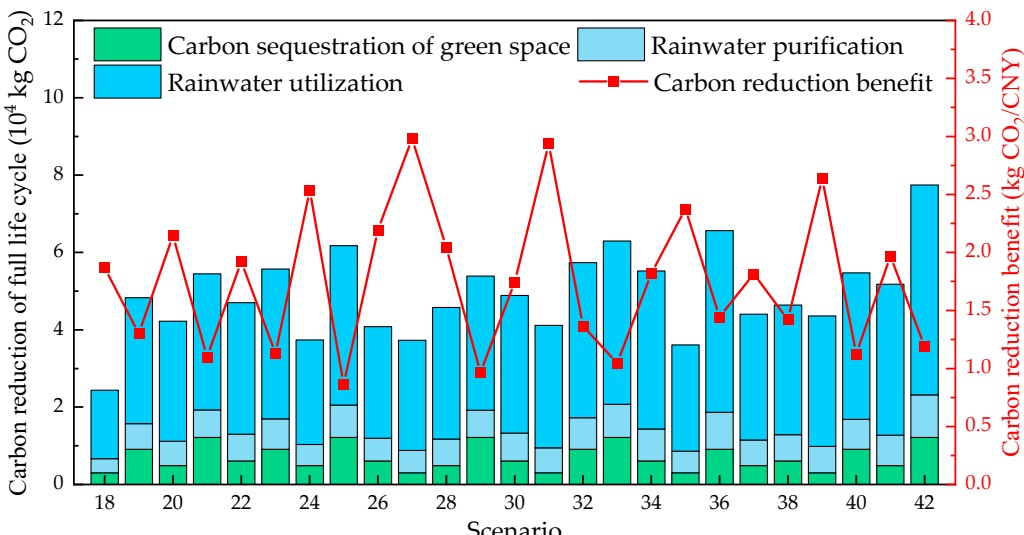

**Figure 11.** Results of scenarios 19–42 for carbon reduction benefit.

**Table 5.** Orthogonal experiment results for carbon reduction benefits.

|     | CC | AH | PC | FA |
| --- | --- | --- | --- | --- |
| K1 | 8.3336 | 7.5071 | 7.9616 | 12.8031 |
| K2 | 9.7038 | 9.2546 | 8.7986 | 10.5100 |
| K3 | 9.0567 | 8.2575 | 9.0417 | 9.0959 |
| K4 | 8.4990 | 9.6396 | 8.9138 | 6.3631 |
| K5 | 8.3436 | 9.2779 | 9.2210 | 5.1646 |
| R | 0.4690 | 0.2700 | 0.4160 | 1.5020 |

### 3.5. Analysis of the Relationship between VCRA and Carbon Emission Intensity

The *VCRA* and carbon intensity for scenarios 1–42 were analyzed by categorizing the test results into 11 groups based on preopening and closing intervals. The intervals were determined by starting from 45% and progressing in 5% increments. The data were then analyzed in Figure 12 using box-and-whisker plots.

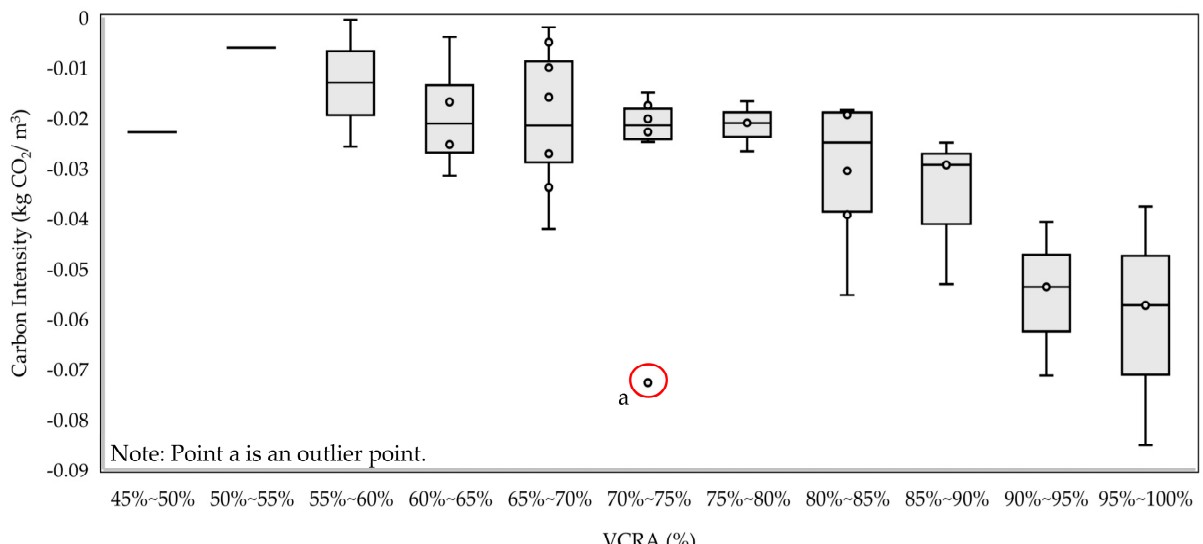

**Figure 12.** Carbon intensity under different levels of *VCRA*.

As depicted in Figure 12, the distribution of *VCRA* varied widely, spanning from 45% to 100%. Data points within different intervals of *VCRA* exhibited a non-uniform

distribution, with the intervals of 65% to 70% and 80% to 85% having the largest number of distributions, each containing eight test scenarios. Within the 65% to 70% interval, scenario 17 exhibited the minimum carbon intensity at $-0.0852$ kgCO$_2$/m$^3$, while scenario 39 reached the maximum carbon intensity of $-0.0005$ kgCO$_2$/m$^3$. In the 80% to 85% interval, carbon intensity increased from $-0.0553$ kgCO$_2$/m$^3$ for scenario 33 to $-0.0185$ kgCO$_2$/m$^3$ for scenario 12. The intervals with the fewest scenarios were 45~50% and 50~55%, each with only one test scenario. The outlier point (in Figure 12, point a) is attributed to the constancy of carbon emissions during the two major phases of construction and demolition when the area of the bioretention facility remains constant. The impact of other factors influencing stormwater retention capacity on the carbon intensity gradually becomes more apparent. This is particularly notable in scenarios where the infiltration coefficient undergoes orders of magnitude-sized changes, leading to a reduction in the amount of stormwater treatment. The dual factor changes result in a noticeable effect on carbon intensity. Therefore, this point is disregarded in the analysis.

These findings suggest two key points. Firstly, certain bioretention facility scenarios can effectively capture rainwater volume within a specified range. Some scenarios meet local requirements, with the *VCRA* exceeding 80% and eliminating the need for additional LID facilities in the study area. Secondly, within the same *VCRA* interval, carbon intensity values varied significantly among schemes, indicating that different scenarios with the same range yield different carbon emission effects. Therefore, studying how to minimize carbon intensity while ensuring the *VCRA* is a direction of research that is worthy of exploration. The carbon intensity exhibits a decreasing trend with the increase in the *VCRA*. As the deployment area significantly influences carbon intensity throughout the entire life cycle, the schemes in the 75~80% interval have the same deployment area, resulting in a stable carbon emission intensity. The interval of 60~65% experiences the largest fluctuation in carbon intensity, with scenario 25 demonstrating the maximum value at 182 units. Kaykhosravi et al. [61] demonstrated that BFs were ranked highest for their social, economic, and environmental benefits, with environmental benefits centered around factors such as the amount of pollutants treated. This study also revealed that bioretention facilities exhibit a greater potential in terms of carbon intensity. In this interval, the infiltration coefficient and aquifer height are at lower values, highlighting the significant impact of the deployment area on the carbon emission intensity [33].

*3.6. Relationship between FLCC and Bioretention Facility Performance in Terms of Carbon Emissions*

A comparative analysis of the size and cost of bioretention facilities and all of their capacities in different scenarios was carried out. The results of the analysis are shown in Figure 13. Figure 13a illustrates that the water control capacity of the BF gradually increases with the rise in cost, leading to a simultaneous decrease in the full life carbon emission and carbon emission intensity. The primary driver for the increased cost is the expansion of the facility area. The figure illustrates a direct proportional relationship between the size of the BF and the cost. The relationship between the cost and the full life cycle carbon emissions is inverse, and it can be learned from Figure 13b,c that if the area of the bioretention facility deployment increases, the volume of water that can be captured by the facility during rainfall increases, the area of green space can lead to carbon sinks, and the amount of carbon sequestered increases with the increase in the area of deployment. These carbon sinks effectively offset the heightened carbon emissions incurred during the construction, operation, and maintenance of bioretention facilities. Han et al. [62] indicate that the larger the size of the BF, the higher the social benefits. However, their findings did not include results related to carbon emissions within the context of social benefits. Furthermore, the change in cost and carbon intensity displayed an inverse proportionality, indicating that the carbon emissions per unit of water controlled by bioretention facilities decreased as the deployment area increased.

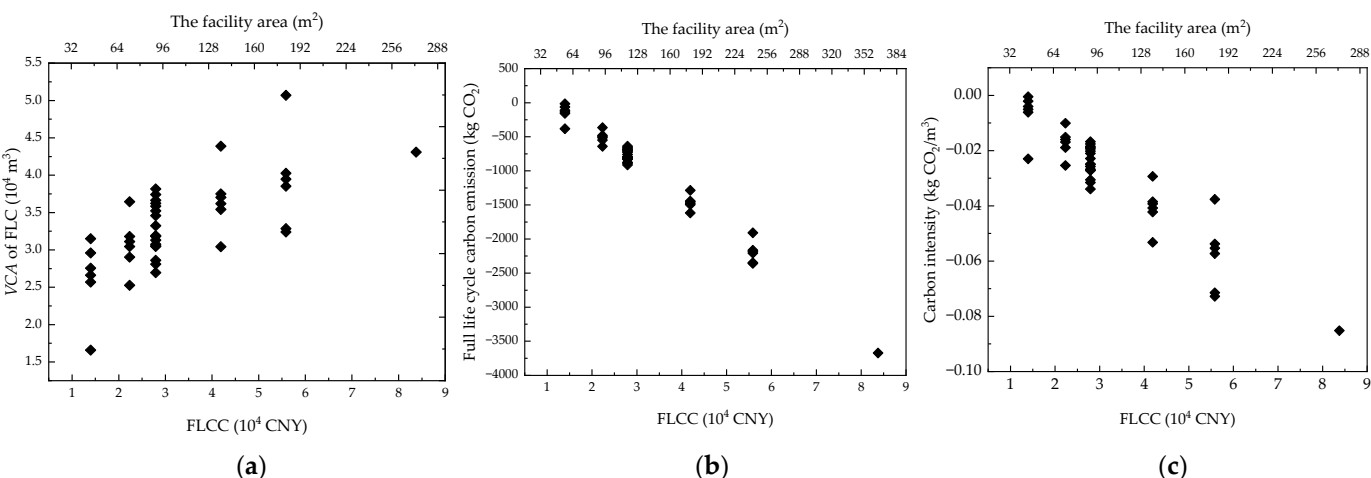

**Figure 13.** FLCC and (**a**) water control capacity, (**b**) carbon emission, and (**c**) carbon intensity.

*3.7. Relationship between Carbon Emission Intensity and Carbon Emission Reduction*

In order to further explore the relationship between carbon emission intensity and carbon reduction benefits, this study conducted a linear fitting of the two quantities for all scenarios, and the results are shown in Figure 14.

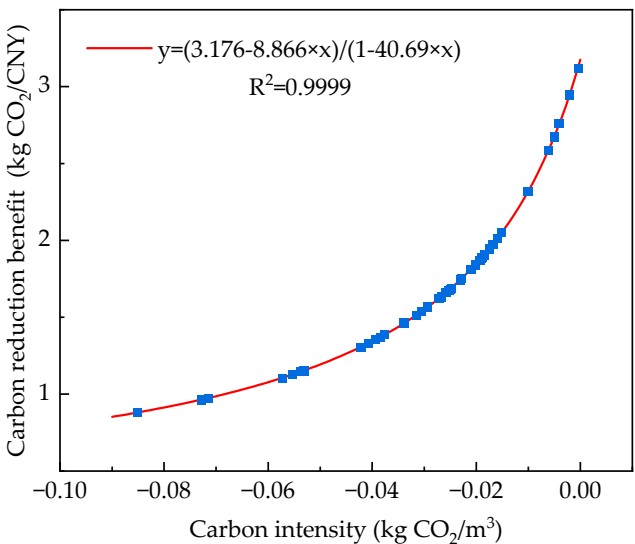

**Figure 14.** The relationship between carbon emission intensity and carbon emission reduction benefits.

As illustrated in Figure 14, there is a well-established fitting relationship between carbon intensity and carbon reduction benefit. This relationship is attributed to the higher proportion of costs that are associated with increased volume capture of rainfall per unit area, indicating a positive correlation between water volume control and carbon intensity. As carbon emissions increased, the carbon abatement benefits of bioretention facilities improved. These findings align with the results reported by Su et al. [33] in the literature. Therefore, stronger carbon abatement methods and emphasizing an intensified approach are recommended for bioretention facilities with greater carbon emission intensity. Attention should be directed towards optimizing carbon abatement effectiveness in these scenarios.

## 4. Conclusions and Prospects

### 4.1. Conclusions

In this study, the carbon emissions of bioretention facilities in different scenarios were calculated using the full life cycle model, and the carbon intensity of bioretention facilities in the full life cycle was evaluated using the carbon intensity of volume capture of rainfall. The influencing factors were analyzed to refine scenarios and optimize facilities. The results are shown in the following:

(1) The carbon intensity of the volume capture of rainfall effectively assesses the carbon emission levels of bioretention facilities, providing a theoretical foundation for the study of carbon emissions in sponge cities.

(2) The carbon intensity value ranges from a maximum of $-0.0005$ kg $CO_2/m^3$ to a minimum of $-0.0852$ kg $CO_2/m^3$, exhibiting a significant difference of approximately 169 times. This value is not only affected by the external environmental changes, but also by the bioretention facility's own attributes such as the aquifer height, permeability coefficient, and facility area.

(3) The results of orthogonal experiments show that the strongest influence on the carbon intensity of the volume capture of rainfall is the facility area, with a correlation coefficient of 0.0520. Under the consideration of the total runoff reduction effect and the carbon emission situation, the bioretention facilities can be prioritized by adjusting the deployment area to satisfy the requirements of the deployment.

(4) The maximum carbon reduction benefit of bioretention facilities is 3.1223 kg $CO_2/CNY$, differing approximately 2.55 times from the minimum value of 0.8802 kg $CO_2/CNY$. For bioretention facilities with a higher carbon emission intensity, emphasis should be placed on carbon emission reduction efforts, and various initiatives can be implemented to enhance their carbon reduction benefits.

### 4.2. Prospects

This study has focused on a singular bioretention facility, necessitating additional research to substantiate the carbon intensity of LID facilities. The research directions encompass:

(1) Investigating the varied impact of different LID facilities on rainwater control, prompting further exploration of carbon intensity for individual LID facilities.

(2) Conducting a study on the carbon intensity of combined LID arrangements at the parcel level, capitalizing on their synergistic effect in enhancing rainfall and flood control.

(3) Climate conditions exert a significant influence on stormwater runoff capture at LID facilities. Employing more accurate climate prediction methods can facilitate research on the carbon emission intensity across various climate conditions.

(4) Constructing a carbon emission model for LID facilities based on data from prior studies. The model will consider varying climatic conditions, utilizing the total runoff control rate as a target. This exploration aims to unveil the potential for carbon emission reduction and strategies for sponge city construction.

**Author Contributions:** Conceptualization, B.W.; Methodology, B.W. and D.W.; Formal Analysis, D.W. and H.C.; Investigation, H.C., W.L. and L.C.; Data Curation, X.L. and H.L.; Writing—Original Draft Preparation, D.W.; Writing—Review and Editing, T.Z.; Supervision, X.W. and J.L. All authors have read and agreed to the published version of the manuscript.

**Funding:** The National Natural Science Foundation of China (No. 52060014); National Key Research and Development Program of China (No. 2018YFE0206200); Gansu Province Construction Science and Technology Project (No. JK2023-18); Gansu Science and Technology Plan (No. 21JR7RA340).

**Data Availability Statement:** Data are available upon reasonable request to the corresponding author.

**Conflicts of Interest:** Author Chen Hai was employed by the company Tianshui Housing and Urban-Rural Development Bureau. Authors Li Wei, Cao Lianbao, and Liu Jianlin were employed by the company Beijing General Municipal Engineering Design & Research Institute Co., Ltd. The

remaining authors declare that the research was conducted in the absence of any commercial or financial relationships that could be construed as a potential conflict of interest.

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
