# Peer review of "The Carbon Emission Intensity of Rainwater Bioretention Facilities"

_water, doi:10.3390/w16010183_

Round 1

Reviewer 1 Report

Comments and Suggestions for Authors

It is an interesting paper, and the authors have done significant work on it. However, several issues should be addressed before publication.

1. More information about urban development should be provided in the introduction section. I suggest the authors read and cite the following papers if appropriate.

          Exploring the temporal and spatial effects of city size on regional economic integration: Evidence from the Yangtze River Economic Belt in China.

          How urban sprawl influences eco-environmental quality: Empirical research in China by using the Spatial Durbin model

2. The font in Figure 1 should be adjusted to improve the readability.

3. The authors did a lot of work on the results, which is good. However, I suggest the authors separate the results section and discussion section.

4. In the discussion section, the authors need to compare their results with other studies, and highlight their contributions and limitations in this field.

5. In the results section, the authors just demonstrated their results; they did not analyze the underlying reasons why the results can be observed.

6. There are so many figures throughout the text. I recommend that the authors condense the content, in order to improve the cohesion and coherence of the manuscript.

Reviewer 2 Report

Comments and Suggestions for Authors

Overall, your manuscript presents a topic of significant importance but requires major revisions for clarity, depth, and scientific rigour. Addressing these specific points will greatly enhance the impact and credibility of your research.

  • 1. Expand on how bioretention facilities specifically contribute to carbon emission reduction. Reference recent studies that investigate similar urban environmental challenges. This will provide a stronger rationale for your study.
  • 2. Critically evaluate recent literature focusing on the effectiveness of bioretention systems in different urban contexts. Compare these findings with your assumptions and hypotheses.
  • 3. Clarify why Tianshui, China, was chosen as the study location and how its specific climatic conditions might influence the results. Provide a more detailed rationale for selecting the variables like aquifer height and permeability coefficient.
  • 4. Elaborate on the statistical models used for analyzing carbon emission intensity. Discuss why these models are appropriate for your data type and how they compare with other possible analytical methods.
  • 5. The results section lacks detailed statistical data presentation. Include more tables and figures that illustrate the relationship between bioretention facility area and carbon emission intensity. Discuss any anomalies in the data.
  • 6. Deepen the discussion by linking your findings to the broader implications for sustainable urban planning. Compare your results with those of similar studies and discuss any discrepancies or similarities.
  • 7. Strengthen your conclusions by directly tying them to the evidence presented. Discuss the practical applications of your research in urban planning and policy-making, emphasizing how your findings can contribute to the development of more sustainable cities.
  • 8. Broaden the scope of your references to include more recent and diverse studies. This will not only validate your research but also position it within the current scientific discourse.
Comments on the Quality of English Language

The quality of English language in the manuscript requires moderate editing. The overall structure and meaning are generally clear, but there are instances of grammatical, syntactical, and idiomatic issues that impact readability and clarity. Attention to these aspects would significantly enhance the manuscript's comprehensibility and professional presentation. A thorough review for language improvements is recommended.

Round 2

Reviewer 2 Report

Comments and Suggestions for Authors

The revisions the authors have made adequately address the concerns raised during the initial review process, and the manuscript now presents a thorough and insightful analysis of the subject matter. Therefore, I recommend accepting the paper for publication.